# The Effect of Climate Variability on Cultivated Crops' Yield and Farm Income in Chiang Mai Province, Thailand

Yadanar Kyaw [1], Thi Phuoc Lai Nguyen [1,*], Ekbordin Winijkul [2], Wenchao Xue [2] and Salvatore G. P. Virdis [3]

1 Department of Development and Sustainability, School of Environment, Resources and Development, Asian Institute of Technology, Pathumthani 12120, Thailand
2 Department of Energy, Environment, and Climate Change, School of Engineering and Technology, Asian Institute of Technology, Pathumthani 12120, Thailand
3 Department of Information and Communication Technology, School of Engineering and Technology, Asian Institute of Technology, Pathumthani 12120, Thailand
* Correspondence: phuoclai@ait.asia

**Abstract:** Agriculture, entwined with climatic conditions, plays a pivotal role in Thailand's sustenance and economy. This study aimed to examine the trends of climate variability and its correlation with crop yields and social and farm factors affecting farm net income in Chiang Mai province, Thailand. Time series climate data (2002–2020) on temperature and rainfall and yields were analyzed using the Mann–Kendall trend test and Sen's slope estimation to investigate the trends and their changes. The Pearson correlation was used to assess the association between climate variability and cultivated crop yields, and multiple linear regression was used to detect the factors influencing the farm net income. The findings show that the total annual rainfall showed an unchanged trend, but the annual temperature increased over time. Higher temperature negatively impacted longan yield but positively affected maize, with no significant impact on rice yield. The rainfall trend had no effect on crop yields. Despite declining trends in some cultivated crops' yield, farm net income was unaffected by individual crop types. Farm income relied on cumulative output and geographic location. This research emphasizes the need for integrating climate data and forecasting models considering agronomic and socio-economic factors and crop suitability assessments for specific regions into adaptation policies and practice.

**Keywords:** temperature; rainfall; rice (*Oryza sativa* L.); maize (*Zea mays* L.); longan (*Dimocarpus longan* L.); farm net income; total farm output volume; production costs; labor investment

## 1. Introduction

When a climate feature, like temperature or precipitation, deviates from the average, this is called climate variability [1]. Extreme weather or climate occurrences are inevitably brought on by climate and weather variability. Unusual events such as heatwaves (high temperature), cold waves (low temperature), downpours (heavy precipitation), and droughts (low precipitation) are a few examples of these phenomena that are more exceptional and severe than regular or ordinary weather [2]. According to the IPCC 2012 [3], global shifts in climate variability and extremes have resulted in a transition of the entire climate distribution toward warmer conditions, signaling a change in the average climate state. This change means that we should expect to see more hot weather, including high temperatures that break records, while seeing less cold weather, including low temperatures that break records. However, in the future, we should expect a rise in both hot and cold weather events, encompassing records for extreme heat and cold [4].

Climate variability is already exerting significant influences on biological systems and the livelihoods of smallholders and the communities reliant on them. The growing trends of climate variability and extreme events pose substantial concerns for agriculture [5]. This is because agricultural yields and production levels exhibit considerable fluctuations

in response to shifts in temperature and precipitation. It is crucial to emphasize that climate stands as the foremost determinant impacting agricultural yield [6]. Variability in temperature and rainfall is estimated to account for a substantial 30–50 percent of the annual variations observed in global cereal yields [7].

Agriculture, intricately intertwined with climatic conditions, stands as a keystone of Thailand's sustenance and economy. However, the formidable force of climate change has begun to significantly impact this delicate balance, particularly evident in Thailand. The altering climate patterns have instigated substantial shifts in agricultural yields and production, necessitating a reevaluation of traditional farming practices [8–10]. Statistics reveal that Thailand's average temperature has increased by approximately 1.04 °C over 40 years from 1970 to 2009, leading to more frequent and intense droughts and floods [11,12]. Precipitation variations, driven by El Niño Southern Oscillation, have led to fluctuations, including record-low rainfall in 2019 (1343.4 mm) and record-high rainfall in 2017 (2017 mm), within 2015–2021 [13,14]. These climatic upheavals have prompted farmers to reconsider their crop choices [15] and planting calendars [16] and change farming systems [17], adapting to the changing climate anomalies. This complex interplay between climate change and agriculture has also reverberated through policy corridors, compelling the Royal Thai Government to restructure agricultural policies to bolster resilience against these challenges [18,19]. With nearly 40% of Thailand's population engaged in agriculture [20], the consequences are vast, making it imperative to harness innovation, scientific insights, and adaptive strategies to ensure food security and the well-being of its agricultural communities in the face of this evolving climate scenario.

Thailand's favorable climate, a tropical climate influenced by seasonal monsoon winds bringing a stream of warm moist air from the Indian Ocean causing abundant rain over the country, makes it an ideal place for cultivating several crops and tropical fruits, making it one of the world's richest sources of these tropical fruits and a global leader in rice exporters [21,22]. Agricultural sectors in Thailand are characterized by a diversity of crops cultivated on local farms. These encompass a range of cereals such as rice (*Oryza sativa* L.), maize (*Zea mays* L.), and wheat (*Triticum aestivum* L.), along with an array of fruits like longan (*Dimocarpus longan* L.), lychees (*Litchi chinensis* Sonn.), mangoes (*Mangifera indica* L.), and tangerines (*Citrus reticulata*). In addition, the cultivation of vegetables, oilseeds like palms, soybeans, and sunflowers, and fiber crops like cotton further contribute to the multifaceted agricultural mosaic [23]. In this context, rice stands as a pivotal economic driver within the agricultural landscapes of Southeast Asia, with Vietnam and Thailand emerging as the foremost nations in rice cultivation [24,25]. Additionally, Thai fruits are popular among global consumers, boasting a remarkable diversity of over 1000 varieties of tropical and sub-tropical fruits in Thailand. Among these, 57 types of fruit are commercially produced on approximately 1.2 million hectares of land [22,26]. As per recent findings by the Statista Research Department (2022), the agricultural sector made a substantial economic contribution of around THB 1.38 trillion (equivalent to approximately USD 38 billion) to Thailand's economy. The crop cultivation sectors played a pivotal role in driving this economic impact, registering the highest contribution among all other sectors [27]. Crop cultivation is adaptable to various scales, spanning from modest endeavors like family farms or community gardens to intermediate setups such as partially commercial family farms and further to expansive operations like commercial family farms or commercial estates [28]. Climate change's impacts can have positive and negative effects on agriculture, presenting significant challenges for farmers and their production [29]. Moreover, another study found that rainfall and La Niña events positively influenced cassava cultivation, while climate variability and extreme events played roles in yield [30]. In the case of rubber, an average temperature above 28 °C negatively affected the growth rate, and higher rainfall caused a decrease in tapping days and latex yield [31]. The concerning consequences of climate change are the altering distribution and abundance of pests and diseases, creating threats to crops and leading to reduced yields [32]. Moreover, climate change is also affecting water availability, leading to increased water scarcity in various

regions, thereby impacting irrigation systems and limiting water accessibility for crops. Thailand is experiencing water shortages due to shifting precipitation patterns and climate change that lead to changes in water availability, resulting in increased water scarcity in various regions, impacting irrigation systems and reducing water availability for crops [33].

Chiang Mai stands as the third most significant contributor to Thailand's GDP, playing a pivotal role in agricultural advancement through its commitment to sustainable farming methods [34]. However, Chiang Mai has witnessed numerous climate change-driven anomalies in agroclimatic conditions, including periods of drought, instances of flooding, occurrences of landslides, and even wildfires. These events have had significant repercussions on its agricultural sector, rural economy, and the ways of life of its inhabitants [35]. During the period from 2006 to 2016, the northern district of Chiang Mai province experienced climate changes that led to pronounced agricultural challenges, particularly water scarcity due to a prolonged period of dry weather. This made it difficult for rice farmers as it caused more diseases and pests, dried out the soil, made it infertile, and resulted in lower rice yields [36]. On the other hand, by using species distribution models (SDMs), it has been projected that numerous species within Chiang Mai province face the risk of extinction by 2050 and 2080, primarily due to the effects of climate change. Notably, a significant portion of the local population, including certain ethnic groups in Chiang Mai, depends on the cultivation of approximately 400 medicinal plant species as a source of income. Therefore, it is imperative to raise climate change awareness among the Karen people and promote sustainable practices in the harvesting and utilization of these medicinal plants to safeguard both the plant biodiversity and the livelihoods of these communities [37].

Furthermore, farmers also encounter numerous challenges to boosting productivity, although they transitioned from traditional to modern farming practices expecting better results [38,39]. This is due to the fact that farmers encounter numerous challenges related to managing a variety of crops and engaging in multiple farming activities, including issues with farm size, increased workload and higher investment requirements, limited knowledge and expertise, restricted market access, environmental risks, and resource allocation [39–44]. Overall, certain socio-demographic conditions of farmers and the impact of climate variability can provide both positive and negative impacts on the performance of agriculture. Thus, this study aimed to assess the impact of climate trends in Chiang Mai province, Thailand, on the yields of key cultivated crops and to investigate how changes in crop yields might influence farm income or any other socio-economic variables that affect it. The goal of this study is to provide valuable insights for agricultural policymakers and extensionists, enabling them to formulate appropriate agricultural policies and practices that consider not only climate factors but also other crucial aspects of the agricultural sector.

## 2. Materials and Methods

### 2.1. Study Area

Chiang Mai's landscape predominantly consists of forests and is situated in the northern part of Thailand, along the Ping River, with geographical coordinates of 18°42′21.83″ northern latitude and 98°58′54.178″ east longitude. The average elevation is 700 m above sea level, with temperatures ranging from 11 °C to 39 °C and an average annual rainfall of around 100 mm. It covers a total area of 20,107 square km, divided into forest (14,060 sqKm), agricultural (2937 sqKm), and non-agricultural/residential areas (3470 sqKm) [34,45]. In terms of farming, there are various systems employed across the total agricultural land area with 134,625 landholders. Chiang Mai province comprises 25 districts (Figure 1) with a population of 1.69 million as of 2015, comprising 818,100 males and 864,064 females engaged in the practices of horticulture, raising livestock, and cultivating aquaculture [46].

Rice (*Oryza sativa* L.), maize (*Zea mays* L.), and longan (*Dimocarpus longan* L.), selected as examined crops in this study, stand out as the predominant crops in relation to their harvested area, production volume, and yields per hectare. Chiang Mai is the third major rice-growing region in northern Thailand, serving as a crucial household staple in the ASEAN community, where per capita consumption is projected to increase around 1% annually [47,48]. Maize, a key

crop in northern Thailand, meets local needs due to population growth and a thriving livestock industry, being highly lucrative and in demand both domestically and internationally [45,49]. Longan, a key agricultural commodity in Chiang Mai, played a pivotal role in elevating Thailand's position as the world's second-largest longan producer [50,51].

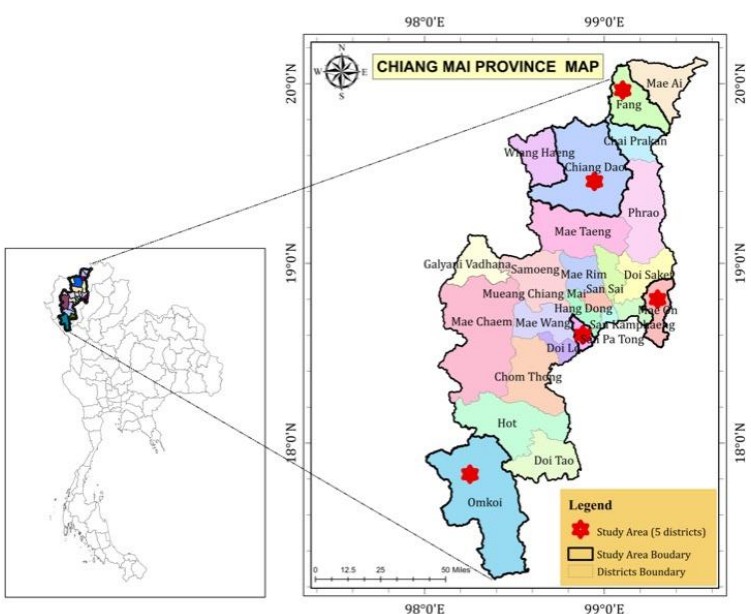

**Figure 1.** Study area.

*2.2. Data Types*

2.2.1. Secondary Data

The details of the secondary data types and sources are presented in Table 1.

**Table 1.** Secondary data for this study.

| | Types of Data | Materials | Scale | Sources |
|---|---|---|---|---|
| ● | The number of farming households in Chiang Mai | Chiang Mai agricultural census, 2013 | District Level | National Statistical Office, Ministry of Information, and Communication Technology |
| ● | The statistical data on cultivation and yield in Chiang Mai | Statistic reports | District Level | Website of relevant government organizations<br>(1) Department of Agricultural Extension Chiang Mai http://www.chiangmai.doae.go.th/web2020/ (Accessed on 25 March 2023)<br>(2) Chiang Mai Provincial Livestock Office https://pvlo-cmi.dld.go.th/webnew/index.php/th/ (Accessed on 25 March 2023)<br>(3) Chiangmai Fisheries Provincial Office https://www4.fisheries.go.th/fpo-chiangmai (Accessed on 25 March 2023) |

**Table 1.** *Cont.*

| Types of Data | Materials | Scale | Sources |
|---|---|---|---|
| ● Climate data<br> - Temperature (T)<br> - Rainfall (R) | Time series climate data: monthly mean of T and R (2002–2020) | Provincial Level Station Data | Thai Meteorological Department—https://www.tmd.go.th/ (Accessed on 1 April 2023) Thaiwater—https://tiwrm.hii.or.th/v3/ (Accessed on 1 April 2023) |

### 2.2.2. Primary Data

Primary data were collected from 259 households from five selected districts of Chiang Dao, Fang, Mae On, Omkoi, and San Pa Tong. The questionnaire survey was conducted between 2022 and 2023 focusing on farm cultivation types, yield, income, and socio-economic factors affecting farm income.

The allocation of sample sizes is proportionately distributed, with representations of Chiang Dao (27%), Fang (23.9%), San Pha Tong (17.4%), Omkoi (20.1%), and a smaller 11.6% for the Mae On district. As depicted in Table 2, a noteworthy finding emerges: 70.3% of farming household heads identify as male, contrasting with the 29.7% representing the female demographic. This disparity underscores the dominant presence of males within the farming community. Shifting the focus to educational achievements, outcomes indicate that a significant portion of respondents have completed their primary education, accounting for 69.5%, while 15% of respondents achieved high school education and secondary education levels, respectively. Age composition forms another salient parameter explored in this research. In total, 49% of respondents fall within the working age range (31 to 60 years). Notably, an aging population within the farming community can be considered to account for 44.8% of the seasoned labor force. Turning to farming experience, close to 41% of the surveyed individuals possess less than two decades of agricultural experience. Another segment, comprising a similar portion of 41.3%, boasts farming experience ranging from 21 to 40 years. The remaining respondents boast over 40 years of farming experience.

**Table 2.** General information on respondents in the study area (N = 259).

| Variables | Number (%) |
|---|---|
| District | |
| Chiang Dao | 70 (27%) |
| Fang | 62 (23.9%) |
| Mae On | 30 (11.6%) |
| Omkoi | 52 (20.1%) |
| San Pa Tong | 45 (17.4%) |
| Gender | |
| Female | 77 (29.7%) |
| Male | 182 (70.3%) |
| Age | |
| Under 30 years | 16 (6.2%) |
| 31 years to 60 years | 127 (49%) |
| Over 60 years | 116 (44.8%) |
| Education | |
| Primary | 180 (69.5%) |
| Secondary | 39 (15.1%) |
| High | 40 (15.4%) |
| Farming Experience | |
| Under 20 years | 108 (41.7%) |
| 21 years to 40 years | 107 (41.3%) |
| Above 40 years | 44 (17%) |

### 2.3. Data Analysis

This research employed an integrated approach, incorporating diverse sources and types of data. Figure 2 provides a detailed depiction of the research design framework employed in this study.

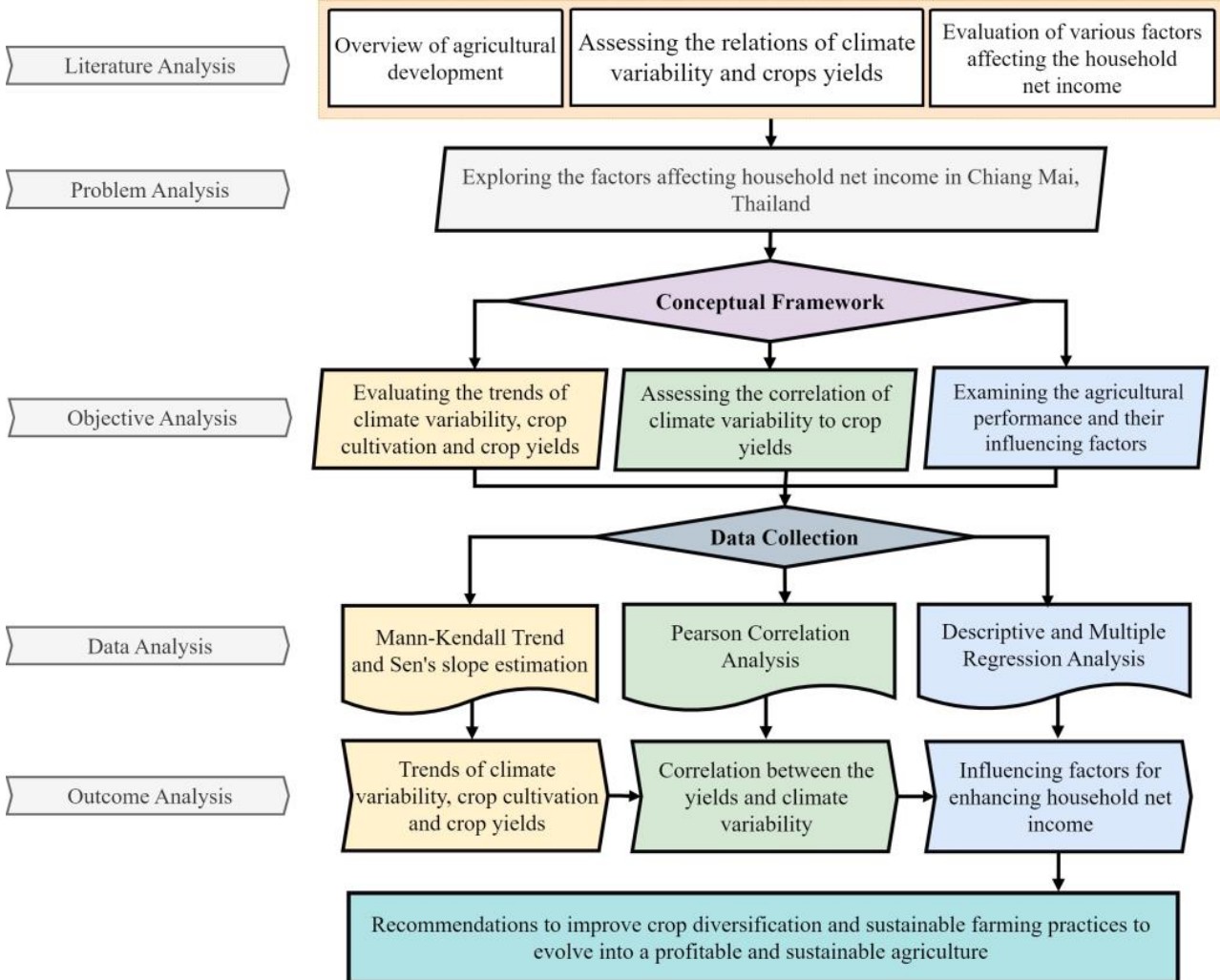

**Figure 2.** Methodology framework.

### 2.3.1. Analysis of Long-Term Climate Variability and Trends

The assessment of temporal trends of climate variability involved the application of the Mann–Kendall trend analysis and Sen's slope estimation techniques, utilizing time series climate data on monthly mean temperature and rainfall across a two-decade time span (2002 to 2020) and from two meteorological stations: station ID No. 327202 (Doi Ang Khang), located in the northern part, and station ID No. 327501 (Mueang Chiang Mai) situated in the middle part of Chiang Mai province. The Mann–Kendall trend, a non-parametric statistical test, was used to assess the presence of a monotonic trend in rainfall, temperature, crop area, and crop yield [52,53]. This method is frequently employed in the fields of climate science and environmental research to analyze extensive datasets spanning extended periods, including variables such as temperature, precipitation, river flow, floods, and others [54]. The computation of the Mann–Kendall test for a time series

involves employing the following equations to ascertain the presence of an upward or downward trend [55].

$$sgn\ (x_i - x_j) = \begin{cases} 1, & x_i - x_j > 0 \\ 0, & x_i - x_j = 0, \\ -1, & x_i - x_j < 0 \end{cases} \tag{1}$$

$$E[S] = \sum_{i=1}^{n=1} \sum_{j=i+1}^{n} sgn(x_i - x_j) \tag{2}$$

$$VAR\ (S) = \frac{1}{18} \{n(n-1)(2n+5) - \sum_{j=1}^{p} t_j(t_j - 1)(2t_j + 5)\} \tag{3}$$

$$Z = \begin{cases} \frac{E[S]-1}{var(s)} & S > 0 \\ 0 & if\ S = 0 \\ \frac{E[S]+1}{var(s)} & S < 0 \end{cases} \tag{4}$$

In Equations (1) and (2), $x_i$ and $x_j$ represent the value of the time series at a particular position, $i$ and $j$. $S$ in Equation (2) represents the sum of the signs of the differences between pairs of observations. "$n$" represents the total number of observations in the time series, and $t$ represents the Kendall rank correlation coefficient, which measures the strength and direction of the monotonic trend in the time series.

Sen's slope estimator is a non-parametric method used to estimate the slope or trend of a time series dataset and is commonly employed in various fields, including hydrology, climatology, and environmental sciences [55,56]. To estimate the magnitude of the temporal trend over historical data, the following equation was applied [57].

$$\beta = Median\ \left(\frac{x_j - x_i}{j - i}\right),\ j > i \tag{5}$$

In Equation (5), $\beta$ represents Sen's estimated slope, $\beta > 0$ means an upward trend, and $\beta < 0$ is a downward trend in a time series. Additionally, $x_i$ and $x_j$ are the values of the time series at positions $i$ and $j$, respectively, and $(j - i)$ is the time interval between the two observations.

Moving to examine the correlation between climate variability and crop yield, the Person correlation analysis equation was applied to measure the strength and direction of the linear relationship.

$$r = \frac{\sum\left((x_i - \bar{x})(y_i - \bar{y})\right)}{\left(\sqrt{\left(\sum\left((x_i - \bar{x})^2\right)\right)}\right)\left(\sqrt{\left(\sum\left((y_i - \bar{y})^2\right)\right)}\right)} \tag{6}$$

$x_i$ and $y_i$ from Equation (6) represent the individual values of the two variables, while $\bar{x}$ and $\bar{y}$ represent the means of the two variables, respectively. The correlation coefficient ($r$) is calculated to assess the statistical significance of the correlation, which can be determined by examining the confidence interval (*p*-value). The interpretation of the Pearson correlation analysis allows us to understand the presence of positive and negative relationships between climate factors and crop yield. The historical temperature and rainfall data are derived from annual records, which could potentially lead to reduced accuracy when assessing the correlation with seasonal crop yields. Nonetheless, longan cultivation is a year-round endeavor and is influenced by temperature and rainfall across all seasons. On the other hand, maize and rice are cultivated twice a year. Thus, from monthly climate data, we calculated the annual average for both temperature and rainfall to perform correlation analysis with the crop yields.

### 2.3.2. Analysis of Socio-Economic Factors Influencing Farm Net Income

Multiple linear regression analysis was utilized to find the association of multiple independent variables (socio-economic) with one dependent variable (farm income). The model is presented as follows:

$$Y = \alpha + b_1 x_1 + b_2 x_2 + \cdots + b_n x_n \tag{7}$$

where $Y$ is the dependent variable; $\alpha$ is the constant; $b_1, b_2, \ldots, b_n$ are the beta coefficients for independent variables; and $x_1, x_2, \ldots, x_n$ are the independent variables (see Table 3).

**Table 3.** Description of variables for analysis (N = 259).

| Variables | Description of Variables | Types of Variables |
|---|---|---|
| Districts | Chiang Dao district: baseline (dummy = 0) Fang district = 1, 0 = otherwise Mae On district = 1, 0 = otherwise Omkoi district = 1, 0 = otherwise San Pa Tong district = 1, 0 = otherwise | Nominal |
| Farmers' age (years) | | Continuous |
| Farmers' gender | Male = 1, female = 0 | Nominal |
| Farmers' education level | | Ordinal |
| Labor working experience (yrs) | | Continuous |
| Labor working days | Number of working days of labor in the farm per year | Continuous |
| Household member | Number of members in household | Continuous |
| Total farm laborers | Number of total laborers on farms (both family labor and rented labor) | Continuous |
| Farm size | Size of farm (ha) | Continuous |
| Total output | Farm output of specific crop (kg) | Continuous |
| Soil types | Soil Information of respondent' district | Nominal |
| Location (lat and long) | Latitude and longitude of respondents | Continuous |
| Crop types | Other crops: baseline (dummy = 0) Rice = 1, otherwise= 0 Maize = 1, otherwise = 0 Longan = 1, otherwise = 0 | Nominal |

## 3. Results

### 3.1. Long-Term Trend of Climate Variability in Chiang Mai (between 1990 and 2020)

Figure 3 shows the results of trend analysis using the Mann–Kendall trend test and Sen's slope. The analysis revealed a significant increasing trend in annual temperature (Kendall's tau = 0.696, $p < 0.0001$, and Sen's slope = 0.07 °C/year) over the studied period (1990–2020) with an increment of 0.07 °C for every individual year. This indicates that the temperature has been steadily rising over time, especially in 1998, 2005, 2015, and 2019. On the other hand, the analysis showed that annual rainfall has remained relatively stable throughout the same period (Kendall's tau = 0.018, $\beta$ = 1.5 mm per year, $p$-value = 0.916). A high fluctuation in rainfall over time was observed, especially a high increase in 1994, 2002, 2006, 2011, 2017, and 2020 and a decrease in 2003, 2012, 2013, and 2018. This suggests that annual rainfall amounts have significant changes in some years but without a notable increase or decrease in long-term trends.

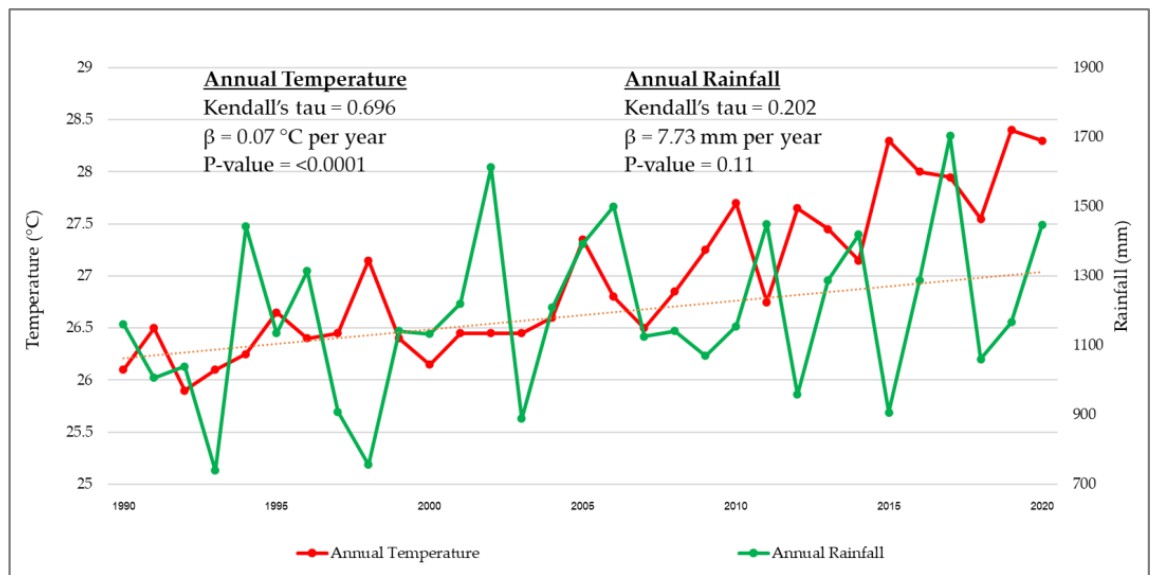

**Figure 3.** Trends of climate variability in terms of average rainfall and temperature between 1990 and 2020 in Chiang Mai province.

### 3.2. Trend of Main Crop Yield in Chiang Mai between 2002 and 2020

The high and significant value of Kendall's tau (0.649) and Sen's Slope (β) of 6122 kg/ha/year and the low *p*-value (*p* < 0.0001) demonstrate a steady increase in maize yield over the studied period. Rice yield showed a notable negative trend (Kendall's tau = −0.404, β = −5503 kg/ha/yr, *p*-value = 0.01), suggesting a decline in rice yield over time. Longan yield also shows a significantly negative trend (Kendall's tau = −0.591, β = −3298 kg/ha/yr, *p*-value = 0.0000), indicating a substantial decline in yield (Figure 4).

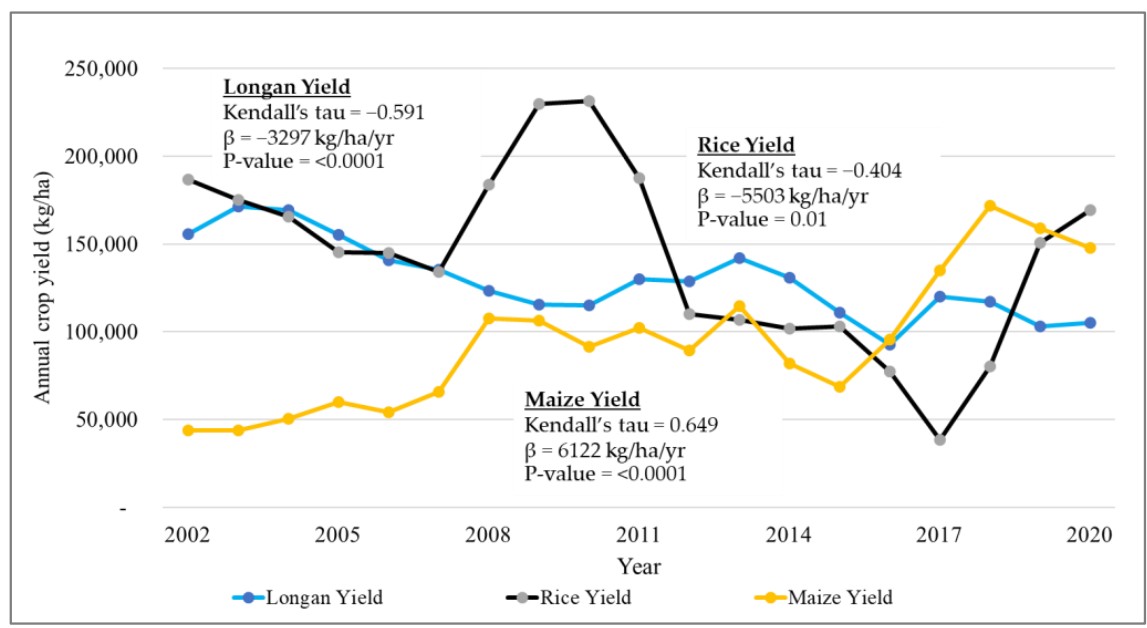

**Figure 4.** Temporal trends of crop yields between 2002 and 2020 in Chiang Mai province.

### 3.3. Correlation between Climate Variability (Rainfall and Temperature) and Crop Yield

Table 4 presents the correlation coefficients (R) and *p*-values obtained from the Pearson correlation analysis of the yields of rice, maize, and longan and temperature and rainfall.

**Table 4.** Correlation between climate variability and crop yield.

| Variables | Coefficient | Longan Yield | Rice Yield | Maize Yield |
|---|---|---|---|---|
| Annual temperature | R | **−0.78** | −0.38 | **0.64** |
| | *p*-Value | **0.000** | 0.11 | **0.003** |
| Annual rainfall | R | 0.048 | −0.15 | 0.04 |
| | *p*-Value | 0.85 | 0.53 | 0.84 |

The correlation analysis reveals that the computed correlation coefficient (R) stands at 0.64, suggesting a positive correlation between maize yield and average annual temperature. This signifies that as the average annual temperature rises, there tends to be an increase in maize yield. The *p*-value of 0.003 underscores the statistical significance of this correlation, indicating that the observed connection is unlikely to occur randomly. The correlation coefficient (R) between longan yield and average annual temperature is −0.78. It indicates an inverse relationship, implying that as the average annual temperature increases, the longan yield tends to decrease. The *p*-value of 0.000 suggests that this correlation is highly statistically significant, and the observed relationship is unlikely to be due to random chance. In the case of rice, the correlation coefficient (R) is −0.38, a negative correlation, implying that higher average annual temperatures are associated with lower rice yields. However, the *p*-value of 0.11 indicates that this correlation is not statistically significant at 0.05. Hence, it cannot be claimed that there is a significant link between rice yield and average annual temperature based on these data. Regarding the correlation between average annual rainfall and the three crops' yield, the high *p*-values (maize = 0.84, longan = 0.85, and rice = 0.53) indicate that this correlation is not statistically significant.

*3.4. Socio-Economic Factors Influencing Net Income*

This section demonstrates the impact of various socio-demographic factors, farm factors, and agricultural outputs on the outcome variable (farm net income) for farmers. The findings of the multiple regression are summarized in Table 5, in terms of regression coefficients (β) and associated *p*-values for each of the independent variables.

**Table 5.** Socio-economic factors influencing farm income.

| Variables | Influencing Factors on Net Income | |
|---|---|---|
| | Coef (β) | *p*-Value |
| (Constant) | | 0.163 |
| Chiang Dao district (baseline) | | |
| Fang district | 0.149 | 0.005 |
| Mae On district | −0.007 | 0.890 |
| Omkoi district | −0.038 | 0.643 |
| San Pa Tong district | −0.006 | 0.914 |
| Education level | 0.051 | 0.291 |
| Age of household head | 0.047 | 0.523 |
| Female (HH gender) | 0.011 | 0.810 |
| Farming experience of household head (years) | −0.008 | 0.899 |
| Number of households members engaged in farming | 0.044 | 0.323 |
| Total laborers on farm | 0.030 | 0.532 |
| Labor working days | −0.251 | 0.000 |
| Total farm size (Ha) | 0.041 | 0.378 |
| Total yield (kg/Ha) | −0.072 | 0.114 |
| Total output volume (kg) | 1.098 | 0.000 |
| Total input cost | −0.629 | 0.000 |
| Other crops (baseline) | | |
| Rice | 0.050 | 0.381 |
| Maize | 0.045 | 0.341 |
| Longan | 0.099 | 0.048 |

Note: Dependent variable: farm net income. R = 0.775 and R-square = 0.601.

A very good correlation coefficient of R = 0.775 expressed a strong relationship between the examined variables. Moreover, approximately 60.1% of the variability in household net income can be accounted for by the independent variables in the model. From this model, it is evident that several independent variables play a noteworthy role in influencing household net income. Notably, among the five districts, the Fang district exhibits (β = 0.149 and *p*-value of 0.015) a significant impact on net income. Likewise, the total output volume of crops (kg) exercises a considerable influence, as evidenced by its coefficient (β) of 1.089 and *p*-value of 0.000. Conversely, the total cost per year displays a negative impact, with a coefficient (β) of −0.624 and a *p*-value of 0.000, indicating that increased costs are linked to reduced net income. Similarly, the total labor working days per year showcase a negative relationship with net income, depicted by its coefficient (β) of −0.251 and *p*-value of 0.000. However, variables such as age of household head (years), gender, educational level, farming experience of household head (years), number of household members, number of household members engaged in farming, total farm size (ha), total productivity per year (kg/ha), farming practices, number of laborers engaged in farming, and crop types were found to be not statistically significant in influencing farm net income based on their non-significant *p*-values.

## 4. Discussion

The results obtained from the analysis of data spanning from 2002 to 2020 present valuable insights into the changing climate patterns and their potential impact on agricultural areas and crop yields. The findings from this study reveal a significant increase in annual temperatures over the years, while annual rainfall has remained relatively stable. The climatic trends in Chiang Mai are found to be similar to those in other locations in the lower Mekong region, such as Kampong Cham province, Cambodia [58], and Ho Chi Minh city, Vietnam [59]. These climate trends may have profound implications for agricultural practices and crop production.

Regarding rice yield, differing from other studies in other regions, such as the Yangtze River Valley, China [60], annual temperature and rainfall were not found to be significantly associated with it. However, an overall decreasing trend can be seen. In the year 2008, the introduction of Japonica rice cultivation was initiated, demonstrating the capacity to deliver noteworthy yields during both dry and wet climatic conditions [61]. This resulted in an increase in the rice plantation in 2008. However, the trend of rice plantation areas significantly decreased after 2008 due to the low yield of rice. This pattern can be attributed to the recurrent droughts faced by Thailand on an annual basis, with a notably impactful instance occurring in the year 2010. Moreover, a separate study posited that the decline in rice yield was exacerbated by the extended drought spanning from 2015 to 2017 [62]. The consequences of the drought in 2010 were particularly terrible for rice cultivators in the northern and northeastern regions of Thailand. These areas observed the emergence of brown planthopper (BPH) infestations, a pest capable of causing severe detriment to rice crops [63].

Thailand encountered a significant natural calamity in 2011—the ninth-largest flood on record. The widespread impact of this event extended to infrastructural damage and the devastation of farmlands and irrigation systems [64]. The consequential aftermath, encompassing the time frame of 2011 to 2012, saw a diminished economic performance in the rice sector. Climatic instabilities like the tropical storm Sock-Ten and the 2011 flood contributed to this downturn [29,64]. It is noteworthy that upland rice cultivation in northern Thailand is beset by a constellation of limitations. Predominant among these are drought, soil erosion, weeds, shallow root structures in rice plants, and the presence of chemical residues in the soil. These factors collectively exert more pronounced constraints on yield compared to the relatively lesser impact of pests and diseases [65].

In the investigation of the relationship between longan yield and temperature, the increased annual temperature in Chiang Mai has yielded negative results in longan yield. Consequently, the trend of longan yield has decreased significantly in the last decades.

Since the rainfall trend was found to be stable in Chiang Mai, no significant correlation was found between rainfall and longan yield in the province, whereas it has had a negative effect on longan yield in Lampang [66]. Although the yield of longan was found to be decreased, the plantation areas were shown as stable with a slight increase in the last 5 years. This is because longan is a long-term fruit tree suitable for planting in some specific geopedological conditions of Chaing Mai. Studies conducted in the Chiang Mai and Lumpum provinces from 1983 to 2009 have shown that longan yield is optimal when minimum temperatures range from 16.27 to 17.47 °C; in contrast, rainfall exceeding 63 mm leads to reduced yield [65]. However, extreme temperatures and excessive rainfall during the flowering period can cause flower drop, potentially affecting pollination and fruit set [67]. In Chiang Mai, there are high temperatures of more than 35 °C from February to July, and extreme temperatures during this period are a major constraint for the flowering and ovary growth of the longan fruit [68]. Additionally, research has suggested that longan trees may be vulnerable to drought during critical stages such as flowering and early fruit development [69]. Thailand faced a devastating drought from November 2009 to August 2010, with temperatures exceeding 40 °C; in addition, between 2015 and 2016, one of the worst droughts in decades caused critically low water levels in reservoirs, mainly affecting central and northeastern areas [70].

In terms of maize, the production of maize demonstrated a noteworthy and favorable connection with the temperature trend. The increased yield was found to be positively associated with the higher annual temperature. As a result, an increased trend of maize yield was found, and consequently, the increased trend of maize plantation areas was also observed over the last decades. Although both maize yield and plantation areas have shown increased trends, historically, approximately 20% of the cultivated land for maize experienced significant damage due to drought-related stress, leading to substantial reductions in yield. To tackle this challenge, the Royal Thai Government introduced the "Maize Breeding Program" with the objective of creating varieties that exhibit resilience to drought conditions [71]. Furthermore, the "Strategic Plan for Maize (2012–2017)" was formulated to boost productivity and reinforce climate adaptability [71]. According to a different scholar, it was argued that maize does not exhibit strong responsiveness to variations in temperature and rainfall.

However, the types of agricultural systems and agronomic practices have a significant impact on how climate variability affects the yields of cultivated crops. The rainfed agricultural system is more vulnerable than the irrigated system to the negative effects of climatic variability because it completely depends on the frequency, intensity, and timing of rainfall [72].

Our study revealed that the trend in maize yield is not correlated with precipitation patterns but rather with temperature. This is because maize production in our region primarily relies on irrigated systems. In contrast, rainfed maize production in Ethiopia [73] and the US Midwest [74] has shown a negative correlation with climate variability, particularly in response to rising temperatures and unstable precipitation. A range of agronomic factors have the potential to counteract the impacts of climate variability on crop yields. Among these, agronomic practices related to the planting calendar, such as earlier planting of existing cultivars and the adoption of improved cultivars, hold promise. Early planting can leverage the warming temperatures at the start of the growing season and help avoid late-season heat stress. Improved cultivars, characterized by new traits, extended growth periods, and more efficient kernel development, exhibit greater yield potential and may serve as a buffer against the effects of climate change. Additionally, enhancing various agronomic aspects, including fertilizer application, the use of certified seeds, and extension services, can contribute to improved outcomes in agriculture [75]. Therefore, agricultural modeling that takes into account various climate scenarios, farm conditions, and socioeconomic factors could assist farmers in adopting appropriate adaptation practices to address climate variability.

The results also revealed that while the yields of Chiang Mai's primary crops, rice and longan, exhibited declining trends, the overall farm net income remained unaffected by the performance of individual crops' yield or crop types. The farms' net income appeared to rely on the cumulative output volume garnered from the entire farm. In response to the diminished yields, with the support of the Royal Thai Government through agricultural policies and incentives, farmers in this study area demonstrated a proactive approach by altering their crop selections and integrating a variety of crops within their farms. This strategic diversification ensured that the total harvested volume remained sustainable, allowing one crop to compensate for the shortcomings of another [76]. The findings also showed that geographic locations affect the farms' net income. In this case, Fang district had a better net income than other districts. Indeed, farms operating in less favored areas often struggle to achieve sufficient profits [77]. Nonetheless, the farms' net income faces a detrimental impact due to escalated production costs and increased labor investment. An explanation for this could be that intensive agriculture fails to yield improved financial returns for the farmers. Moreover, it is important to note that policy changes, market dynamics, and technological innovations can also have an impact on crop yields and farm net income, even though these variables were not included in our analysis. For instance, it is worth highlighting that the Royal Thai government has placed a significant emphasis on promoting the adoption of organic farming for sustainable agriculture over the last two decades. The transition to organic farming may initially lead to yield declines in the first few years, but over time, it can result in increased yields and the ability to command better market prices.

## 5. Conclusions

This study portrays a comprehensive view on climate variability in Chiang Mai, Thailand, along with the yield trends of main cultivated crops (rice, maize, and longan). While this study was centered on a specific location in Chiang Mai, its findings could have the same implications for other similar regions in Southeast Asia that share similar climate conditions, cropping systems, and socio-economic factors. The research findings confirm that, like in other Southeast Asian regions, increasing temperatures have both positive and negative effects on crops, benefiting certain varieties (such as maize in this case) while negatively impacting others (like longan). Nevertheless, the results also indicate declining trends in the yield of Chiang Mai's primary crops, rice and longan. Remarkably, the overall farms' net income remains unaffected by the yield performance of individual crops or crop types. Instead, the farms' net income seems contingent upon the cumulative output volume generated across entire farms. This research strongly recommends that agricultural policymakers and extensionists incorporate climate data and forecasting models into their adaptation strategies. It also suggests optimizing resource management, improving agricultural practices, diversifying crops, and assessing crop suitability in specific regions. Additionally, this study reveals that heavy investments in inputs and labor negatively affect farm income, suggesting that promoting sustainable agriculture is wiser than intensive farming.

Furthermore, this research emphasizes the challenges faced by farms in less favorable areas, like Omkoi, which have a high concentration of ethnic minority groups with limited access to knowledge, technology, and markets. This study urges a focus on these disadvantaged districts, implementing targeted policies to enhance social capital and build resilience against climate change and natural disasters.

There are some limitations of this research that must be acknowledged. Firstly, the accuracy and reliability of our analysis might be influenced by the availability of historical data. However, while it is uncommon to find comprehensive, long-term time series climate data in Southeast Asia, the 19-year time series climate data we had access to for the Chiangmai area hold significant value in depicting regional climate trends across a period of two decades. Secondly, some agronomic practices and socio-economic factors, such as policy changes, market dynamics, and technological innovations, were not included in

the correlation and regression model, which might overlook the nuanced relationships between crop yield, farm income, and other factors. For future research, it is suggested to integrate crop modeling that considers various climate scenarios, farm conditions, and socio-economic factors. Incorporating semi-structured interviews to capture farmers' perspectives and local knowledge [78,79] in order to suggest desirable adaptation measures at the farm level is recommended.

**Author Contributions:** Conceptualization, T.P.L.N.; methodology, Y.K., T.P.L.N. and S.G.P.V.; validation, T.P.L.N.; formal analysis, Y.K.; investigation, T.P.L.N.; data curation, Y.K., T.P.L.N. and S.G.P.V.; writing—original draft preparation, Y.K. and T.P.L.N.; writing—review and editing, Y.K., T.P.L.N., E.W., W.X. and S.G.P.V.; visualization, Y.K.; supervision, T.P.L.N.; funding acquisition, T.P.L.N., E.W., W.X. and S.G.P.V. All authors have read and agreed to the published version of the manuscript.

**Funding:** This research was carried out within the project "Application of Big Earth Data in Support of the Sustainable Development Goals in Thailand" funded by the National Research Council of Thailand (NRCT).

**Institutional Review Board Statement:** The study was conducted according to the guidelines of the Declaration of Helsinki and approved by the Research Ethics Review Committee of Asian Institute of Technology (AIT) (Ref. No.: RERC 2022/015 on 14 September 2022).

**Informed Consent Statement:** Informed consent was obtained from all farmers who participated in our survey.

**Data Availability Statement:** The data presented in this study are available on request from the corresponding author.

**Acknowledgments:** The authors acknowledge the data support from the Land Development Department, Thai Meteorological Department and National Statistical Office, Ministry of Information and Communication Technology. The authors would like to thank Jureerut Somboon for her significant contribution to the primary data collection in this study and the members under the SDG2 component of the project "Earth Observation Big Data for SDGs" funded by NRCT Thailand.

**Conflicts of Interest:** The authors declare no conflict of interest.

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
