# Peer review of "The Effect of Climate Variability on Cultivated Crops’ Yield and Farm Income in Chiang Mai Province, Thailand"

_climate, doi:10.3390/cli11100204_

Round 1
Reviewer 1 Report
I carried out the review on the paper “Assessment of the effects of climate variability on crop yield and socioeconomic factors on farm income: A case study in Chiang Mai province, Thailand”. My comments are divided in to two types: general and specific ones.
General comments:
One thing I see is that, the research was not carried out for the whole country, it was just for Chiang Mai, however in the introduction the information authors provide with is about Thailand and Thai. I think the information in introduction should be in the context of the regions studied in this work.
About the time series analysis, it is some confuse since that in the abstract authors say that was carried out just for the period 2002-2020, that is to say, 19 years. Also, it is not clearly mentioned how many weather stations were used, or is the climate data a regional average?
Specific comments:
Title. Too long, 24 words. Authors should try to compact it.
Introduction. Lines 30-31. First line of introduction, this information it is not too relevant for showing it with a reference, since it is also almost the same problem in the whole globe. However, the information you provide is important, but please share it without reference.
Introduction. In the first paragraph, authors start directly with the description of Thailand climate situation. It is recommended first, start with the global situation of climate change, please do this within an additional paragraph (it will be the first one of introduction). Also, I would suggest to add a second paragraph, in this paragraph would be important to talk about changes on temperature and precipitation in specific regions (countries) of the world; this because in the paper climate change is mentioned many times and also climate variability is mentioned in the title, in this paragraph, would be very nice to have a definition of climate change/climate variability.
Introduction. Lines 48-79. In some lines of this paragraph, authors say “Thai”, perhaps, it is a local form of referring to Thailand. I would suggest just to use official names of the country. Also, in this paragraph author use the currency “baht”, perhaps, would be better to use a more common or universal currency (maybe USD) to understand money quantities without any transformation. After this paragraph, is necessary to add another paragraph in which authors should talk about the approach they are using in this study, I mean they should mention other works in which economic and climate correlations were used to know the influence of climate variability in economic factors such as incomes.
Line 88. Authors say “0f”, perhaps they wanted to say “of”.
At the end of introduction. Authors should state clearly the objective of the paper.
Study area section. All the content of this section should be moved to material and methods section; within this late section, please add the name of the subsection “study area”. This section looks very extensive, please summarize. Also in this part, authors mention some problems of the study area, they could move this information to introduction to highlight a little more the problems in Thailand.
Materials and method section. It is not mentioned how many weather stations were used. If you are using an average of all weather stations please explain. In Table 1, you say the period was 30 years but in the abstract, you say the period 2002-2020, which is not 30 years and this is the minimal for climate change analysis purposes. This is very important given that in the first paragraph of discussion, you highlight a changing (positive) temperature pattern. It is very important to say how many stations, given that the main part of the paper is the temperature and precipitation analysis.
Results section. Lines 236-247. Are the trends you mention here for a specific site or are they a regional average? When a change in climate is supported by statistical numbers, so we can say *statistically significant trend*, I see in some cases you say "insignificant", you should say "not significant" or "not statistically significant".
Results section. Line 239 and 240. Please use the correct units for °C.
Results. 4.2 section. Honestly, I do not find this information be essential for presenting it. The increase or decrease of a cultivation area can be seen/consulted in the official statistics of the country.
Results section. Authors should add references in this section. It is important to know if similar/different findings were obtained regarding other studies. Were your trends similar/different to other ones reported for your study area? How do they behave regarding to the trend range reported for the whole world? Please use references.
Discussion section. I would have liked to see more agronomic explanation regarding to the changes in yield and its connection to temperature/precipitation. There are a lot of papers in modelling about the effect of temperature and precipitation on yield.
I hope these comments help the authors to prepare a strengthened version of the paper.
Author Response
R: I carried out the review on the paper “Assessment of the effects of climate variability on crop yield and socioeconomic factors on farm income: A case study in Chiang Mai province, Thailand”. My comments are divided into two types: general and specific ones.
A: We are grateful for the careful review work and constructive comments from the Reviewer which helped us improve significantly our manuscript. We have revised carefully the manuscript based on the comments and suggestion of the reviewer as follows:
General comments:
R: One thing I see is that, the research was not carried out for the whole country, it was just for Chiang Mai, however in the introduction the information authors provide with is about Thailand and Thai. I think the information in introduction should be in the context of the regions studied in this work.
A: Thank you very much for this suggestion. The study area of Chiangmai has been introduced in the introduction of the paper (Line 80- 136)
R: About the time series analysis, it is some confuse since that in the abstract authors say that was carried out just for the period 2002-2020, that is to say, 19 years. Also, it is not clearly mentioned how many weather stations were used, or is the climate data a regional average?
A: Thank you. We used time series climate monthly data from two meteorological stations. This information has been added to the manuscript (Line 196-199)
Specific comments:
R: Title. Too long, 24 words. Authors should try to compact it.
A: The title has been revised to “The effect of climate variability on cultivated crops yield and farm income in Chiang Mai Province, Thailand”.
R: Introduction. Lines 30-31. First line of introduction, this information it is not too relevant for showing it with a reference, since it is also almost the same problem in the whole globe. However, the information you provide is important, but please share it without reference.
A: We have removed this reference.
R: Introduction. In the first paragraph, authors start directly with the description of Thailand climate situation. It is recommended first, start with the global situation of climate change, please do this within an additional paragraph (it will be the first one of introduction). Also, I would suggest to add a second paragraph, in this paragraph would be important to talk about changes on temperature and precipitation in specific regions (countries) of the world; this because in the paper climate change is mentioned many times and also climate variability is mentioned in the title, in this paragraph, would be very nice to have a definition of climate change/climate variability.
A: Thank you very much for these constructive suggestions, we have now added two paragraphs (Line 30-49) to address the Reviewer’s comment.
R: Introduction. Lines 48-79. In some lines of this paragraph, authors say “Thai”, perhaps, it is a local form of referring to Thailand. I would suggest just to use official names of the country.
A: We have changed the Thai government into the Royal Thai Government, which is the formal translated name of the government of Thailand in the whole manuscript. Whenever the word “Thai” is used as an adjective it is not changed.
R: Also, in this paragraph author use the currency “baht”, perhaps, would be better to use a more common or universal currency (maybe USD) to understand money quantities without any transformation.
A: We have revised the text, adding the equivalent amount in USD.
R: After this paragraph, is necessary to add another paragraph in which authors should talk about the approach they are using in this study, I mean they should mention other works in which economic and climate correlations were used to know the influence of climate variability in economic factors such as incomes.
A: Thank you. A new paragraph has been added as suggested.
R: Line 88. Authors say “0f”, perhaps they wanted to say “of”.
A: Revised.
R: At the end of the introduction. Authors should state clearly the objective of the paper.
A: Thank you. The objective and goal of the study has been added in Line 130 -136
R: Study area section. All the content of this section should be moved to the material and methods section; within this late section, please add the name of the subsection “study area”.
A: We have moved the study area into the material and methods section as suggested.
R: This section looks very extensive, please summarize. Also in this part, authors mention some problems of the study area, they could move this information to introduction to highlight a little more the problems in Thailand.
A: Thank you. This section has been compacted. Some sentences have been moved to introduction to introduce Chiang Mai study area.
R: Materials and method section. It is not mentioned how many weather stations were used. If you are using an average of all weather stations please explain. In Table 1, you say the period was 30 years but in the abstract, you say the period 2002-2020, which is not 30 years and this is the minimal for climate change analysis purposes. This is very important given that in the first paragraph of discussion, you highlight a changing (positive) temperature pattern. It is very important to say how many stations, given that the main part of the paper is the temperature and precipitation analysis.
A: Thank you. They are 19-year monthly climate data (2002-2020) from two stations. All suggested information has been added in the section of material and method section which are highlighted in red color.
R: Results section. Lines 236-247. Are the trends you mention here for a specific site or are they a regional average? When a change in climate is supported by statistical numbers, so we can say *statistically significant trend*, I see in some cases you say "insignificant", you should say "not significant" or "not statistically significant".
A: Thank you. The whole result section has been checked and revised accordingly.
R: Results section. Line 239 and 240. Please use the correct units for °C.
A: Revised.
R: Results. 4.2 section. Honestly, I do not find this information be essential for presenting it. The increase or decrease of a cultivation area can be seen/consulted in the official statistics of the country.
A: Thank you. We do agree that it is not essential to this result, thus the section has been removed.
R: Results section. Authors should add references in this section. It is important to know if similar/different findings were obtained regarding other studies. Were your trends similar/different to other ones reported for your study area? How do they behave regarding to the trend range reported for the whole world? Please use references.
A: Since we have a separate discussion section. All key findings are now discussed or compared with other studies in the discussion section.
R: Discussion section. I would have liked to see more agronomic explanation regarding to the changes in yield and its connection to temperature/precipitation. There are a lot of papers in modelling about the effect of temperature and precipitation on yield.
A: Thank you very much for this suggestion. We have added a new paragraph discussing the results with agronomic explanation and implication for agricultural modeling in Lines 382-403
R: I hope these comments help the authors to prepare a strengthened version of the paper.
A: We appreciate the suggestions and comments of the Reviewer very much. We have carefully addressed all comments and believe the quality of the manuscript has been substantially improved.

Reviewer 2 Report
While the study provides valuable insights into the relationship between climate variability, crop yields, and farm net income in the context of Chiangmai, Thailand, there are a few potential disadvantages or limitations that should be considered:
1. Limited Generalizability: The study focuses solely on a case study in Chiangmai, Thailand. As a result, the findings may have limited generalizability to other regions or countries with different climatic, agricultural, and socio-economic conditions.
2. Data Limitations: The study relies on time-series data from 2002 to 2020 for temperature, rainfall, and crop yields. The availability and quality of historical data could impact the accuracy and reliability of the analysis.
3. Simplified Analysis: The study employs statistical methods such as the Mann Kendal trend test, Sen's slope estimation, and Pearson correlation. While these methods provide valuable insights, they may oversimplify the complex interactions between climate variables and crop yields, potentially overlooking nuanced relationships.
4. Limited Socio-economic Factors: The study focuses on identifying socio-economic factors influencing farm net income, but the scope of factors considered may be limited. Other factors, such as market conditions, labor availability, and technological advancements, could also play a significant role in farm income.
5. Causality and Confounding: While correlation analysis is used to examine relationships, it does not establish causality. There could be confounding variables or other factors not accounted for in the analysis that contribute to observed correlations.
6. Temporal Scale: The study's analysis covers a relatively short time frame (2002-2020). Longer-term trends or shifts in climate patterns and their impact on agriculture may not be fully captured within this timeframe.
7. Farm Diversity: The study focuses on primary crops like longan, maize, and rice. Other agricultural practices, such as livestock farming or diversified cropping systems, may not be adequately represented.
8. Qualitative Aspects: The study primarily employs quantitative methods, potentially missing qualitative insights from farmers' perspectives and local knowledge that could provide a richer understanding of the issues.
9. External Factors: The study does not extensively address external factors such as policy changes, trade dynamics, or technological innovations that could influence crop yields and farm net income.
10. Future Projections: The study focuses on historical trends and correlations. Incorporating future climate projections could enhance the study's relevance in addressing potential climate change impacts.
These limitations, while important to consider, do not negate the value of the study's findings but provide opportunities for further research and deeper exploration of the complex interactions between climate, agriculture, and socio-economic factors.
Line 33-34 could cite the below references to further support the statement.
Liu, K., Harrison, M.T., Yan, H. et al. Silver lining to a climate crisis in multiple prospects for alleviating crop waterlogging under future climates. Nat Commun 14, 765 (2023). https://doi.org/10.1038/s41467-023-36129-4
Malik, A., Li, M., Lenzen, M. et al. Impacts of climate change and extreme weather on food supply chains cascade across sectors and regions in Australia. Nat Food 3, 631–643 (2022). https://doi.org/10.1038/s43016-022-00570-3
Author Response
R: While the study provides valuable insights into the relationship between climate variability, crop yields, and farm net income in the context of Chiangmai, Thailand, there are a few potential disadvantages or limitations that should be considered:
A: Thank you very much for the constructive comments and suggestions. We have carefully revised the manuscript and addressed your comments as follows:
- Limited Generalizability: The study focuses solely on a case study in Chiangmai, Thailand. As a result, the findings may have limited generalizability to other regions or countries with different climatic, agricultural, and socio-economic conditions.
Thank you for this comment. While the study was centered on a specific location in Chiang Mai, its findings could have implications for other similar regions in Southeast Asia that share similar climate conditions, cropping systems, and socio-economic factors. The research findings confirm that, like in other Southeast Asian regions, increasing temperatures have both positive and negative effects on crops, benefiting certain varieties (such as maize in this case) while negatively impacting others (like longan). Therefore, it can be generalized to other similar contexts in the Southeast Asian region.
We have added this point in the paper (Line 429- 434).
- Data Limitations: The study relies on time-series data from 2002 to 2020 for temperature, rainfall, and crop yields. The availability and quality of historical data could impact the accuracy and reliability of the analysis.
Thank you very much. We recognize that the accuracy and reliability of our analysis may be influenced by the availability and quality of historical data. While it's not common to find comprehensive, long-term time series climate data in the Southeast Asian region, the 19-year precise climate data we had access to for the Chiangmai area is already highly valuable in depicting regional climate trends. These findings can serve as a valuable resource for future research aimed at comparing climate variability over time.
This limitation has been added to the conclusion (Line
- Simplified Analysis: The study employs statistical methods such as the Mann Kendal trend test, Sen's slope estimation, and Pearson correlation. While these methods provide valuable insights, they may oversimplify the complex interactions between climate variables and crop yields, potentially overlooking nuanced relationships.
Thank you for this comment. This aspect could be a limitation of the research. We highlighted as limitation in the conclusion section.
- Limited Socio-economic Factors: The study focuses on identifying socio-economic factors influencing farm net income, but the scope of factors considered may be limited. Other factors, such as market conditions, labor availability, and technological advancements, could also play a significant role in farm income.
Thank you. While we acknowledged the limited socio-economic factors included in the regression model such as market conditions, labor availability, and technological advancements, we have discussed these factors in the discussion (see Line 418 – 424)
- Causality and Confounding: While correlation analysis is used to examine relationships, it does not establish causality. There could be confounding variables or other factors not accounted for in the analysis that contribute to observed correlations.
Thank you very much. We do agree with the Reviewer’s comment. The crop yield could be affected by other factors such as agronomic farming practices. Although we did not perform the correlation of crop yields with these factors, we have now highlighted the discussion of these factors in Lines 369-390 and Lines 418-424.
- Temporal Scale: The study's analysis covers a relatively short time frame (2002-2020). Longer-term trends or shifts in climate patterns and their impact on agriculture may not be fully captured within this timeframe.
Thank you. We acknowledge that a 19-year time frame may not provide a complete picture of climate patterns and their effects on agriculture. Nevertheless, despite the absence of data from before the 2000s in this region, it remains worthwhile to conduct this study spanning the last two decades, as this period has witnessed more pronounced and severe climate change as mentioned in the paper.
We consider this as a limitation in the conclusion section.
- Farm Diversity: The study focuses on primary crops like longan, maize, and rice. Other agricultural practices, such as livestock farming or diversified cropping systems, may not be adequately represented.
Thank you for this comment. In this study we focused only on the cultivated crops and longan, maize, and rice are the main crops in this study area. We have highlighted the cultivated crops and justified the choice of these crops in the manuscript under the study area.
- Qualitative Aspects: The study primarily employs quantitative methods, potentially missing qualitative insights from farmers' perspectives and local knowledge that could provide a richer understanding of the issues.
We do agree the paper would have more insights if it included the farmers' perspectives and local knowledge obtained from the semi-structured interviews. Since the sample of the study is quite large, we were unable to interview in-depth farmers. This could be further explored in one of our future research projects in the area.
- External Factors: The study does not extensively address external factors such as policy changes, trade dynamics, or technological innovations that could influence crop yields and farm net income
Thank you. Although we did not include these variables in the analysis, we discussed these factors carefully in Lines 382-403 and Lines 418-424.
- Future Projections: The study focuses on historical trends and correlations. Incorporating future climate projections could enhance the study's relevance in addressing potential climate change impacts.
The scope of this study didn’t include future projections, however as discussed in Lines 398 – 400.
These limitations, while important to consider, do not negate the value of the study's findings but provide opportunities for further research and deeper exploration of the complex interactions between climate, agriculture, and socio-economic factors.
We thank the Reviewer for the constructive comments. We have incorporated all aspects into the discussion and the conclusion section of the manuscript.
Line 33-34 could cite the below references to further support the statement.
Liu, K., Harrison, M.T., Yan, H. et al. Silver lining to a climate crisis in multiple prospects for alleviating crop waterlogging under future climates. Nat Commun 14, 765 (2023). https://doi.org/10.1038/s41467-023-36129-4
Malik, A., Li, M., Lenzen, M. et al. Impacts of climate change and extreme weather on food supply chains cascade across sectors and regions in Australia. Nat Food 3, 631–643 (2022). https://doi.org/10.1038/s43016-022-00570-3
Thank you for these suggestions. Both references have been cited in the paper.

Reviewer 3 Report
In manuscript climate-2575648 “Assessment of the effects of climate variability on crop yield and socioeconomic factors on farm income: A case study in Chiang Mai province, Thailand” authors investigated effect of different aspects of climate change (temperature, precipitation) in Chiang Mai, Thailand, along with the yield trends and plantation areas of its main crops (rice, maize, and longan). Moreover, manuscript also discussed the correlation between temporal changes in crop yield and annual temperature and rainfall, alongside the socio-eco-nomic factors influencing farm net income.
Manuscript is written on one of the main challenges agriculture is facing currently. Although manuscript is fairly written, however, significant improvement required before formal publication.
General Comments
Authors need to focus add details of Climate variability, anomalies and their impact on agricultural crops, which are not sufficiently discussed
English Improvement required
Technical terminologies can be improved for better understanding of concepts discussed in the study
Authors need to clearly mention “Average Annual Temperature” is that annual mean of daily average temperature?
Use standard units e.g., kg and °C. Instead of “Kg” use “kg”
In Discussion, discuss obtained results with the published from Thailand and other countries.
Specific Comments
L-20: Revise the text, what is stable trend?
L-24-25: revise the text
L-27: Revise keywords, remove crop plantation areas
L-35: temperature increase as compared to which base period?
L-41: changes not necessarily disasters, can be anomalies
L-48: describe details of favourable conditions
L-90-91: Improve need of study
L-93: climate-induced effects? Can be “Climate change driven anomalies in agroclimatic conditions…” like increasing temperature, heatwaves, spells of drought and floods
L-248: in Fig. 3. Correct spelling of Annual Temperature
L-344: Rice yield is strongly influenced by temperature and rainfall anomalies. Obtained results can be due to insufficient and inefficient data
L-428: Conclusion is too long. It must be concise and describe key findings and brief recommendation/s
Suggested Citation
Arunrat, N., S. Sereenonchai, W. Chaowiwat and C. Wang, 2022: Climate change impact on major crop yield and water footprint under CMIP6 climate projections in repeated drought and flood areas in Thailand. Science of The Total Environment 807, 150741.
Gadedjisso-Tossou, A., K. I. Adjegan and A. K. M. Kablan, 2021: Rainfall and Temperature Trend Analysis by Mann–Kendall Test and Significance for Rainfed Cereal Yields in Northern Togo. Sci 3, 17.
Rehmani, M. I. A., C. Ding, G. Li, S. T. Ata-Ul-Karim, A. Hadifa, M. A. Bashir, M. Hashem, S. Alamri, F. Al-Zubair and Y. Ding, 2021: Vulnerability of rice production to temperature extremes during rice reproductive stage in Yangtze River Valley, China. Journal of King Saud University - Science 33, 101599.
English editing can significantly improve the manuscript
Author Response
R: In manuscript climate-2575648 “Assessment of the effects of climate variability on crop yield and socioeconomic factors on farm income: A case study in Chiang Mai province, Thailand” authors investigated effect of different aspects of climate change (temperature, precipitation) in Chiang Mai, Thailand, along with the yield trends and plantation areas of its main crops (rice, maize, and longan). Moreover, manuscript also discussed the correlation between temporal changes in crop yield and annual temperature and rainfall, alongside the socio-economic factors influencing farm net income.
Manuscript is written on one of the main challenges agriculture is facing currently. Although manuscript is fairly written, however, significant improvement required before formal publication.
A: Thank you very much for your time and valuable comments which helped us improve our manuscript.
General Comments
- Authors need to focus add details of Climate variability, anomalies and their impact on agricultural crops, which are not sufficiently discussed
Thank you. The two new paragraphs have been added to introduce climate variability, anomalies and their impact on agricultural crops in Lines 30-49.
- English Improvement required
The English of the whole document has been checked and improved.
- Technical terminologies can be improved for better understanding of concepts discussed in the study
Thank you. They are all checked and improved.
- Authors need to clearly mention “Average Annual Temperature” is that annual mean of daily average temperature?
It is the annual temperature which was calculated from the monthly mean temperature of 12 months. All terms have now been corrected in the paper.
- Use standard units e.g., kg and °C. Instead of “Kg” use “kg”
Corrected.
- In Discussion, discuss obtained results with the published from Thailand and other countries.
Thank you. We have enhanced sustainably the discussion which is all highlighted in red color.
Specific Comments
- L-20: Revise the text, what is stable trend?
Changed to “unchanged trend”
- L-24-25: revise the text
Revised.
- L-27: Revise keywords, remove crop plantation areas.
Removed.
- L-35: temperature increase as compared to which base period?
The temperature increase is compared every year over 40 years from 1970 to 2020
- L-41: changes not necessarily disasters, can be anomalies
Changed as suggested.
- L-48: describe details of favourable conditions
Explained Lines 68-70.
- L-90-91: Improve need of study
The need and objectives of the study have been improved. (Lines 103 -123)
- L-93: climate-induced effects? Can be “Climate change driven anomalies in agroclimatic conditions…” like increasing temperature, heatwaves, spells of drought and floods
Changed as suggested.
- L-248: in Fig. 3. Correct spelling of Annual Temperature
Corrected.
- L-344: Rice yield is strongly influenced by temperature and rainfall anomalies. Obtained results can be due to insufficient and inefficient data
We found the rice yield declined in this study but no correlation was found with climate variability. We acknowledged the insufficient and inefficient data in our study. While it's uncommon to find comprehensive, long-term time series climate data in Southeast Asia, the 19-year precise climate data we had access to for the Chiangmai area holds significant value in depicting regional climate trends. These findings can serve as a valuable resource for future research aimed at comparing climate variability over time
We added this limitation in the conclusion.
- L-428: Conclusion is too long. It must be concise and describe key findings and brief recommendation/s
Thank you. The conclusion section has been compacted, however, we have added a paragraph of limitation to accommodate the suggestion of Reviewer 1.
- Suggested Citation
Arunrat, N., S. Sereenonchai, W. Chaowiwat and C. Wang, 2022: Climate change impact on major crop yield and water footprint under CMIP6 climate projections in repeated drought and flood areas in Thailand. Science of The Total Environment 807, 150741.
Gadedjisso-Tossou, A., K. I. Adjegan and A. K. M. Kablan, 2021: Rainfall and Temperature Trend Analysis by Mann–Kendall Test and Significance for Rainfed Cereal Yields in Northern Togo. Sci 3, 17.
Rehmani, M. I. A., C. Ding, G. Li, S. T. Ata-Ul-Karim, A. Hadifa, M. A. Bashir, M. Hashem, S. Alamri, F. Al-Zubair and Y. Ding, 2021: Vulnerability of rice production to temperature extremes during rice reproductive stage in Yangtze River Valley, China. Journal of King Saud University - Science 33, 101599.
Thank you. All suggested references have been cited in the paper.
